# The Multifaceted Role of Macrophages in Biology and Diseases

**DOI:** 10.3390/ijms26052107

**Published:** 2025-02-27

**Authors:** Jan Brancewicz, Natalia Wójcik, Zuzanna Sarnowska, Julia Robak, Magdalena Król

**Affiliations:** Center of Cellular Immunotherapies, Warsaw University of Life Sciences, Building 23, Level 0, Laboratory Number 0135, 8 Ciszewskiego St., 02-786 Warsaw, Poland

**Keywords:** macrophages, autoimmune disease, cancer

## Abstract

Macrophages are highly adaptable immune cells capable of responding dynamically to diverse environmental cues. They are pivotal in maintaining homeostasis, orchestrating immune responses, facilitating tissue repair, and, under certain conditions, contributing to disease pathogenesis. This review delves into the complex biology of macrophages, highlighting their polarization states, roles in autoimmune and inflammatory diseases, involvement in cancer progression, and potential as therapeutic targets. By understanding the context-dependent functional plasticity of macrophages, we can better appreciate their contributions to health and disease, paving the way for innovative therapeutic strategies.

## 1. Introduction

Macrophages, derived from monocytes, are essential components of the innate immune system, ubiquitously present across all tissues. Their primary functions include the phagocytosis of pathogens and apoptotic cells, antigen presentation, and the secretion of a diverse array of cytokines and chemokines. Beyond these roles, macrophages are instrumental in tissue development, homeostasis, and repair. Recent advances in single-cell transcriptomics have significantly refined our understanding of macrophage biology by uncovering extensive heterogeneity, identifying novel subpopulations, and characterizing unique gene expression profiles across different tissues and disease states. This powerful technology has enhanced our appreciation of macrophage functional diversity and their distinct roles in immunity, homeostasis, and pathology, while also highlighting how their inherent plasticity may contribute to diseases such as autoimmune disorders and cancer [1,2].

## 2. Macrophage Polarization and Plasticity

Macrophages exhibit a high degree of plasticity, allowing them to adapt their functional phenotype in response to microenvironmental signals. This adaptability is often described along a spectrum, with the classical M1 and alternative M2 polarization states representing the extremes [3] (Figure 1).

M1 macrophages, induced by interferon-gamma (IFN-γ) and lipopolysaccharide (LPS), are characterized by their pro-inflammatory properties. They produce high levels of cytokines such as tumor necrosis factor-alpha (TNF-α), interleukin-1 beta (IL-1β), and interleukin-6 (IL-6), which are crucial for initiating and sustaining inflammatory responses [4,5]. However, excessive or prolonged M1 activation can lead to chronic inflammation and tissue damage [6,7] (Table 1).

In contrast, M2 macrophages arise in response to signals like interleukin-4 (IL-4) and interleukin-13 (IL-13). These cells are associated with anti-inflammatory functions, tissue repair, and remodeling. M2 macrophages secrete cytokines such as IL-10 and TGF-β, which resolve inflammation and promote tissue healing (Table 1). The M2 phenotype is further subdivided into M2a, M2b, M2c, and M2d, each with distinct functions and marker expressions [7,8] (Figure 1).

It is important to note that macrophage polarization is not a fixed state but rather a dynamic process, resulting in various functional activation states that macrophages can adopt, depending on environmental cues (Figure 1). These activation states encompass not only classical polarization extremes (M1, M2) but also intermediate or specialized phenotypes, reflecting macrophage functional plasticity and adaptability [9]. We highlight several examples of macrophages exhibiting mixed or specialized activation states, shaped by their tissue-specific niches or disease settings (Table 2).

Modern macrophage classifications have shifted from the traditional binary model of activation (M1/M2) to a more nuanced understanding that emphasizes the microenvironment they inhabit and the context of their action. Recent studies have highlighted the existence of hybrid and intermediate macrophage phenotypes that do not fit neatly into this binary model (Table 3). For example, M4 macrophages, induced by platelet factor 4 (CXCL4), exhibit both pro-inflammatory and foam cell-like properties and have been implicated in atherosclerosis [10]. Mox macrophages, which arise under oxidative stress conditions, show distinct transcriptional profiles characterized by NRF2-dependent gene expression and play roles in tissue remodeling [10,11]. M(Hb) macrophages, found in hemorrhagic environments, respond to hemoglobin–haptoglobin complexes and exhibit an iron-recycling phenotype while maintaining some anti-inflammatory characteristics [10]. Mhem macrophages, also associated with hemorrhagic regions, are characterized by high heme oxygenase-1 (HO-1) expression and contribute to plaque stability by promoting cholesterol efflux and reducing lipid accumulation [10,12]. Mres macrophages, known as resolving macrophages, play a crucial role in the resolution phase of inflammation by clearing apoptotic cells and secreting anti-inflammatory mediators, facilitating tissue repair and homeostasis [13]. These specialized macrophage subtypes underscore the complexity of macrophage polarization and emphasize the need to move beyond the classical M1/M2 paradigm toward a more functional and context-dependent classification system. Future research should continue exploring these intermediate states and their relevance in disease pathology and therapeutic targeting.

Understanding the diverse macrophage phenotypes and their functional plasticity provides crucial context for comprehending their varied roles in immune responses and tissue homeostasis. Regardless of their classification, all macrophage subsets share key functional capabilities essential for maintaining organismal health, notably through their involvement in processes such as phagocytosis, efferocytosis, and cytokine secretion. In the following chapter, we delve into these fundamental functions, highlighting how macrophages employ them in both physiological and pathological conditions.

## 3. Phagocytosis, Efferocytosis and Secretory Functions

Phagocytosis, efferocytosis, and secretory functions are interconnected processes central to macrophage biology, reflecting their versatile role in maintaining tissue homeostasis and orchestrating immune responses. These phenomena share a common theme: the ability of macrophages to sense, respond to, and interact with their microenvironment. Through phagocytosis, macrophages clear pathogens and debris, a process closely linked to efferocytosis, where apoptotic cells are efficiently engulfed to prevent inflammation. Simultaneously, their secretory functions allow them to release cytokines, growth factors, and enzymes, influencing immune signaling, tissue repair, and pathogen destruction.

### 3.1. Phagocytosis and Efferocytosis

Phagocytosis and efferocytosis are both essential functions of macrophages, enabling immune surveillance, tissue homeostasis, and the resolution of inflammation. Phagocytosis primarily involves the engulfment and degradation of pathogens, cellular debris, and apoptotic cells. This process begins with the recognition of targets through surface receptors such as pattern recognition receptors (PRRs) and opsonin receptors, which facilitate binding to microbial components or opsonized particles. Upon binding, the target is internalized into a phagosome, which subsequently fuses with lysosomes to form a phagolysosome, where enzymatic degradation occurs. This process is often associated with a pro-inflammatory response, as the degradation of pathogens leads to the release of immune-stimulating molecules that activate additional immune cells [14,15].

Efferocytosis, in contrast, is a specialized form of phagocytosis dedicated to the clearance of apoptotic cells. Unlike the phagocytosis of pathogens, which triggers inflammation, efferocytosis is inherently anti-inflammatory, promoting tissue repair and immune tolerance. The recognition of apoptotic cells involves two distinct mechanisms: direct and indirect PS receptors. Direct receptors, such as BAI1, Tim-4, and Stabilin-2, recognize phosphatidylserine (PS), an “eat-me” signal exposed on the outer membrane of apoptotic cells. In contrast, indirect receptors, including TAM receptors (Tyro3, Axl, and MerTK) and integrins (e.g., αVβ3 and αVβ5), require bridging molecules such as Gas6, Protein S, or MFG-E8 to facilitate apoptotic cell recognition. Among these, BAI1 is particularly notable for its role in activating the Elmo1-Dock180-Rac1 signaling pathway, which is essential for cytoskeletal rearrangements and efficient apoptotic cell engulfment [16]. Once bound, apoptotic cells are internalized in a controlled manner, preventing secondary necrosis and the release of damage-associated molecular patterns (DAMPs), which could otherwise provoke an immune response [15].

A key distinction between phagocytosis and efferocytosis lies in their cytoskeletal regulation, particularly the roles of the small GTPases RhoA and Rac1 (Figure 2). Both processes require dynamic actin remodeling for target internalization, but Rac1 plays a dominant role in efferocytosis, where it drives actin polymerization and membrane extension to facilitate the controlled engulfment of apoptotic cells. In phagocytosis, Rac1 is also involved but to a lesser extent, as pathogen uptake relies more on contractile forces rather than the smooth membrane extensions required in efferocytosis [14]. In contrast, RhoA plays a more significant role in phagocytosis, where it helps generate contractile forces necessary for engulfing rigid structures such as bacteria. In efferocytosis, RhoA acts primarily as a negative regulator, limiting excessive membrane protrusions to ensure apoptotic cells are internalized efficiently without disrupting tissue homeostasis. Dysregulation of this balance, such as insufficient Rac1 activation or excessive RhoA activity, can impair efferocytosis. This impairment leads to apoptotic cell accumulation and secondary necrosis, triggering chronic inflammation associated with diseases like systemic lupus erythematosus and atherosclerosis [14].

Understanding the distinct but overlapping roles of RhoA and Rac1 in phagocytosis and efferocytosis provides critical insights into how macrophages regulate inflammation and tissue homeostasis. The failure of efferocytosis not only disrupts immune resolution but also exacerbates chronic inflammatory conditions by sustaining an environment rich in inflammatory mediators [14,15]. Targeting the molecular regulators of these processes presents a potential therapeutic avenue for autoimmune and inflammatory diseases, aiming to restore effective apoptotic cell clearance and reduce pathological inflammation.

### 3.2. Secretory Functions

Macrophages are prolific secretory cells, producing a wide array of molecules that modulate immune responses and influence other cell types. These secretions include cytokines, chemokines, growth factors, and enzymes. For instance, pro-inflammatory cytokines such as TNF-α and IL-1β drive inflammation, while anti-inflammatory cytokines like IL-10 and TGF-β suppress excessive immune responses and promote tissue repair [17,18].

Through these secretory functions, macrophages not only defend against pathogens but also contribute to tissue remodeling and repair, highlighting their multifaceted roles in maintaining health [19].

The processes of phagocytosis, efferocytosis, and cytokine secretion discussed in Section 3 underscore the central role of macrophages in orchestrating innate immune responses and maintaining tissue homeostasis. These functions also intersect closely with adaptive immunity, as macrophages not only eliminate pathogens and apoptotic cells but also process and present antigens to initiate and regulate adaptive immune responses. In Section 4, we explore macrophages’ critical role in antigen presentation, detailing how they bridge innate and adaptive immunity to achieve comprehensive immune regulation.

## 4. Antigen Presentation and Immune Regulation

Antigen presentation is a fundamental process in the immune system that bridges innate and adaptive immunity. Macrophages, as professional antigen-presenting cells (APCs), play a pivotal role in this process by capturing, processing, and presenting antigens to T cells. Through their interactions with both CD4^+^ helper T cells and CD8^+^ cytotoxic T cells, macrophages influence immune activation, tolerance, and regulation. Their dual role in initiating immune responses and maintaining immune homeostasis underscores their complexity in both health and disease. This chapter explores the mechanisms of antigen presentation, the molecular machinery involved, and the ways macrophages regulate immune responses in various physiological and pathological contexts.

### 4.1. Mechanisms of Antigen Presentation

Macrophages utilize major histocompatibility complex (MHC) molecules to present antigens to T cells, initiating adaptive immune responses. MHC class II molecules present extracellular antigens to CD4^+^ T cells, while MHC class I molecules present intracellular antigens, including those derived from viruses, to CD8^+^ T cells. This dual presentation capacity makes macrophages critical in combating both extracellular pathogens and intracellular infections (Table 4).

The process of antigen presentation begins with the phagocytosis or endocytosis of antigens, followed by their processing within endolysosomal compartments. During this stage, antigens are degraded into peptides, which are loaded onto MHC molecules. For MHC class II, this involves the invariant chain (Ii) that guides MHC molecules to endosomal compartments, where peptide loading occurs. Once loaded, the MHC-peptide complexes are transported to the cell surface, where they engage T-cell receptors (TCRs). This interaction, accompanied by co-stimulatory signals such as CD80 and CD86, determines the activation of naïve T cells and their differentiation into effector T cells [20].

Macrophages also possess the ability to cross-present antigens, a unique process by which extracellular antigens are presented on MHC class I molecules. Cross-presentation is crucial for initiating cytotoxic T-cell responses against tumors and viruses. Recent studies have highlighted the role of endosomal transport proteins, such as Sec22b, in facilitating cross-presentation by directing antigen-containing vesicles to MHC class I processing pathways [21]. This mechanism is particularly important in macrophage subsets within the tumor microenvironment, where effective cross-presentation can shape anti-tumor immunity.

### 4.2. Macrophages in Immune Regulation

Beyond antigen presentation, macrophages play a central role in immune regulation, balancing immune activation with tolerance. Their regulatory functions are mediated through the secretion of cytokines, expression of immune checkpoint molecules, and interactions with regulatory T cells (Tregs).

Pro-inflammatory macrophages, often referred to as M1 macrophages, secrete cytokines such as interleukin-12 (IL-12) and TNF-α, which promote Th1 responses and potentiate inflammation. These cytokines are critical in mounting robust immune responses against intracellular pathogens and cancer. However, excessive activation of macrophages can lead to chronic inflammation and tissue damage, as seen in autoimmune diseases such as rheumatoid arthritis and inflammatory bowel disease [7].

In contrast, anti-inflammatory or M2 macrophages play a pivotal role in immune tolerance and tissue repair. By secreting IL-10 and TGF-β, M2 macrophages suppress effector T-cell activity and promote the expansion of Tregs. These regulatory mechanisms are crucial for preventing autoimmunity and resolving inflammation. In the context of tumors, however, M2-like tumor-associated macrophages (TAMs) exploit these functions to suppress anti-tumor immunity and facilitate immune evasion by cancer cells [19] (see more in Section 7.3).

Checkpoint molecules expressed by macrophages further contribute to immune regulation. For example, PD-L1 interacts with its receptor PD-1 on T cells to inhibit T-cell activation and proliferation. While this mechanism prevents excessive immune responses, it is frequently co-opted by cancer cells to create an immunosuppressive microenvironment. Blocking PD-L1 expression on macrophages, therefore, represents a promising strategy in immunotherapy to reinvigorate exhausted T cells [22].

### 4.3. Antigen Presentation and Immune Regulation in Disease

The interplay between antigen presentation and immune regulation by macrophages is critical in various diseases. In infections, macrophages act as the first line of defense by presenting antigens to T cells and initiating immune responses. However, pathogens such as *Mycobacterium tuberculosis* and human immunodeficiency virus (HIV) can manipulate macrophage antigen presentation to evade immune detection. *M. tuberculosis*, for example, inhibits MHC class II antigen presentation by interfering with endosomal trafficking, thereby allowing the pathogen to persist within macrophages [20] (see more in Section 7.2.1).

In cancer, macrophages within the tumor microenvironment display altered antigen-presenting capabilities. While cross-presentation by macrophages has the potential to activate cytotoxic T cells, tumors often reprogram TAMs to suppress effective antigen presentation. This reprogramming involves the downregulation of MHC molecules and co-stimulatory signals, coupled with the upregulation of inhibitory molecules such as PD-L1. These changes create a tolerogenic environment that hinders anti-tumor immunity and supports tumor growth [23] (see more in Section 7.3).

Autoimmune diseases represent another context where macrophage antigen presentation is dysregulated. In rheumatoid arthritis, macrophages present self-antigens to autoreactive T cells, perpetuating chronic inflammation. Similarly, in systemic lupus erythematosus, defects in macrophage efferocytosis lead to the accumulation of apoptotic cells, providing a source of autoantigens that drive disease progression [24] (see more in Section 7.1).

In conclusion, macrophages play a dual role in antigen presentation and immune regulation, acting as mediators of both immune activation and tolerance. Their ability to process and present antigens, coupled with their regulatory functions, places them at the center of immune responses in health and disease. Beyond immune regulation, macrophages are also pivotal in maintaining tissue integrity through their involvement in wound healing, tissue remodeling, and regeneration processes. In the following chapter, we examine the essential roles macrophages play in tissue remodeling and repair, highlighting both their beneficial actions in physiological regeneration and their detrimental contributions to pathological fibrosis.

## 5. Tissue Remodeling and Repair

Tissue remodeling and repair are critical processes for maintaining tissue homeostasis and recovering from injury, in which macrophages play a central role. As immune sentinels, macrophages coordinate inflammatory responses, clear debris, and secrete mediators that regulate the activity of fibroblasts, endothelial cells, and other immune cells. Their phenotypic plasticity allows macrophages to transition between pro-inflammatory and pro-reparative states, enabling them to balance the resolution of inflammation with the promotion of tissue regeneration. This chapter explores the molecular and cellular mechanisms by which macrophages mediate tissue remodeling and repair, highlighting their role in wound healing, fibrosis, and regeneration.

### 5.1. Macrophages in Wound Healing

Wound healing is a highly coordinated process consisting of three overlapping phases: inflammation, proliferation, and remodeling. Macrophages are key regulators of each phase, transitioning dynamically between functional states to meet the demands of the tissue environment.

During the inflammatory phase, macrophages are recruited to the site of injury through chemokines such as chemokine (C-C motif) ligand 2 (CCL2) and interleukin-8 (CXCL8) [25]. These early-phase macrophages predominantly exhibit an M1-like phenotype, characterized by the production of pro-inflammatory cytokines such as IL-1β, TNF-α, and IL-6. These molecules amplify the inflammatory response, recruit additional immune cells, and combat potential infections [26] (Figure 3). However, persistent M1 activity can lead to excessive inflammation and delayed healing, underscoring the need for a timely transition to a reparative phenotype [25].

As the wound progresses to the proliferative phase, macrophages adopt an M2-like phenotype in response to signals such as IL-4 and IL-13. These M2 macrophages secrete anti-inflammatory cytokines such as IL-10 and TGF-β, which dampen inflammation and promote tissue repair (Figure 3). Additionally, M2 macrophages release growth factors such as VEGF and platelet-derived growth factor (PDGF), which stimulate angiogenesis and the proliferation of fibroblasts. This transition is critical for the deposition of granulation tissue and the formation of new blood vessels, enabling the delivery of oxygen and nutrients to the regenerating tissue [27,28].

In the remodeling phase, macrophages further contribute to tissue maturation by regulating extracellular matrix (ECM) remodeling. They produce matrix metalloproteinases (MMPs) to degrade excessive ECM components, allowing for the replacement of the provisional matrix with organized, functional tissue. Conversely, their secretion of tissue inhibitors of metalloproteinases (TIMPs) ensures that ECM degradation is tightly controlled to prevent tissue instability. This delicate balance between ECM synthesis and degradation highlights the critical role of macrophages in the final stages of wound healing [28,29].

### 5.2. Fibrosis: The Pathological Side of Repair

While macrophages are essential for tissue repair, dysregulated macrophage activity can lead to fibrosis, a pathological condition characterized by excessive ECM deposition and scarring. Fibrosis occurs when the reparative phase of wound healing becomes chronic, often as a result of persistent inflammation or repeated injury.

Macrophages are central to the development of fibrosis due to their secretion of pro-fibrotic mediators such as TGF-β, connective tissue growth factor (CTGF), and PDGF [30]. These molecules drive the activation of fibroblasts into myofibroblasts, which produce large quantities of ECM proteins such as collagen and fibronectin. In chronic liver injury, for example, macrophages promote the activation of hepatic stellate cells, leading to the accumulation of fibrotic tissue and the progression of liver fibrosis [30,31] (see more in Section 7.2.5).

Recent studies have highlighted the heterogeneity of macrophages in fibrotic tissues. Pro-inflammatory macrophages contribute to fibrosis by sustaining inflammation, while pro-reparative macrophages exacerbate ECM deposition through excessive TGF-β secretion. The balance between these subpopulations determines the extent of fibrosis and the potential for tissue recovery [4,30]. Therapeutic strategies aimed at modulating macrophage polarization, such as using colony-stimulating factor 1 receptor (CSF1R) inhibitors, have shown promise in reducing fibrosis by limiting macrophage recruitment and activation [4,31].

Interestingly, some macrophage subsets have been shown to possess anti-fibrotic properties. For example, Ly6C^low^ macrophages in murine models of liver fibrosis secrete MMPs that degrade fibrotic ECM, facilitating tissue remodeling and resolution of fibrosis [32]. Harnessing these macrophage subsets may offer a novel therapeutic avenue for reversing fibrosis in chronic diseases.

### 5.3. Macrophages in Regeneration

In addition to their roles in wound healing and fibrosis, macrophages are key regulators of tissue regeneration in organs such as the liver, heart, and skeletal muscle. Unlike fibrosis, which involves excessive scarring, regeneration restores tissue architecture and function.

Macrophages promote regeneration by creating a supportive microenvironment that facilitates stem cell activation, proliferation, and differentiation. In skeletal muscle, for example, macrophages secrete insulin-like growth factor 1 (IGF-1) and hepatocyte growth factor (HGF) to stimulate satellite cell activation and muscle fiber repair [30]. These regenerative macrophages also clear cellular debris and apoptotic cells through efferocytosis, preventing secondary inflammation that could impede regeneration.

In the liver, macrophages known as Kupffer cells orchestrate the regenerative response following partial hepatectomy or acute liver injury. By releasing growth factors such as HGF and IL-6, they stimulate hepatocyte proliferation and liver tissue regeneration. However, the regenerative capacity of macrophages can be impaired in chronic liver diseases, where persistent inflammation and fibrosis limit their ability to support tissue repair [33].

Cardiac regeneration presents a particularly challenging context for macrophages, as the adult heart has limited regenerative capacity. In zebrafish, which exhibit robust cardiac regeneration, macrophages play an essential role in clearing necrotic tissue and promoting cardiomyocyte proliferation [34]. Translating these findings to mammalian systems may provide new insights into enhancing cardiac repair after myocardial infarction.

Tissue remodeling and repair rely on a delicate balance between inflammation resolution, extracellular matrix deposition, and cellular regeneration. Macrophages not only orchestrate these processes through cytokine secretion and interaction with fibroblasts but also play a critical role in iron homeostasis, which is essential for proper tissue recovery. Iron availability influences macrophage function, as it affects oxidative stress responses, mitochondrial metabolism, and the resolution of inflammation. The ability of macrophages to sequester, store, and release iron distinguishes them from other immune cells and is a key factor in both tissue regeneration and disease pathology. The following section explores the unique iron-handling capabilities of macrophages and their implications for immunity, metabolism, and disease progression.

## 6. Iron Homeostasis and Metabolism in Macrophages

Iron homeostasis is a tightly regulated physiological process critical for maintaining cellular and systemic health. Macrophages play a pivotal role in iron metabolism, acting as key regulators of iron recycling, storage, and distribution. Their ability to sense, uptake, and release iron enables them to influence both local tissue environments and systemic iron levels. Dysregulation of iron metabolism in macrophages contributes to pathological conditions such as anemia of inflammation, atherosclerosis, and infections. This section explores the mechanisms by which macrophages regulate iron homeostasis, their metabolic adaptations, and the implications of macrophage-mediated iron regulation in health and disease.

### 6.1. Mechanisms of Iron Handling in Macrophages

Macrophages are central to systemic iron homeostasis due to their role in recycling iron from senescent erythrocytes. This process, termed erythrophagocytosis, begins with the engulfment of aged or damaged RBCs. Within the macrophage phagolysosome, hemoglobin is degraded, releasing heme, which is then processed by HO-1 into free iron, biliverdin, and carbon monoxide. Free iron is subsequently stored within ferritin, a protein complex that prevents its accumulation in toxic forms, or exported into circulation via ferroportin, the only known mammalian iron exporter [35,36].

The export of iron through ferroportin is regulated by hepcidin, a peptide hormone produced by the liver in response to elevated systemic iron levels or inflammatory signals. Hepcidin binds to ferroportin, inducing its internalization and degradation, which restricts iron efflux from macrophages and other iron-handling cells. This mechanism is crucial for maintaining systemic iron balance but also contributes to the development of anemia of inflammation during chronic diseases. Elevated hepcidin levels in inflammatory states reduce iron availability for erythropoiesis by sequestering iron within macrophages [35,37].

Macrophages also express transferrin receptors (TfR1 and TfR2) to uptake iron bound to transferrin from the extracellular environment. This process is essential for macrophages residing in tissues with high iron turnover, such as the spleen and liver. Additionally, macrophages can internalize non-transferrin-bound iron and heme derived from hemolysis or tissue injury, further underscoring their versatility in iron acquisition [38,39].

### 6.2. Macrophage Iron Metabolism and Its Influence on Immune Function

The metabolic state of macrophages influences their iron-handling capacity and vice versa, with significant implications for immune responses. Iron availability within macrophages modulates their polarization and functional output. Pro-inflammatory M1 macrophages exhibit increased iron sequestration, driven by enhanced ferritin expression and reduced ferroportin activity [35] (Figure 4). This iron retention supports the production of ROS through the Fenton reaction, which enhances their antimicrobial activity. However, this process also contributes to tissue damage during chronic inflammation [39].

In contrast, anti-inflammatory M2 macrophages release stored iron through increased ferroportin expression and decreased ferritin levels (Figure 4). This iron efflux promotes tissue repair and regeneration by providing iron to surrounding cells, such as fibroblasts and endothelial cells, which require iron for DNA synthesis and cellular proliferation. For example, studies in murine models of wound healing have demonstrated that macrophage-derived iron is critical for the proliferation of keratinocytes and the formation of granulation tissue [36,39].

Iron metabolism also plays a role in the macrophage response to infections. Pathogens, such as *Mycobacterium tuberculosis* and *Salmonella enterica*, exploit macrophage iron stores to support their growth [40]. In response, macrophages restrict intracellular iron availability by upregulating ferritin and hepcidin and downregulating ferroportin, a process known as nutritional immunity. This iron withholding limits pathogen replication but may also impair macrophage function and host tissue repair during prolonged infections [37,40].

### 6.3. Implications of Dysregulated Iron Metabolism in Disease

Dysregulated iron metabolism in macrophages is implicated in various pathological conditions, including anemia, atherosclerosis, and neurodegenerative diseases. Anemia of inflammation is a hallmark of chronic inflammatory disorders, where persistent macrophage iron sequestration limits systemic iron availability. Elevated hepcidin levels and reduced ferroportin activity trap iron within macrophages, impairing its release for erythropoiesis. Targeting hepcidin–ferroportin interactions has emerged as a therapeutic strategy for mitigating anemia in inflammatory diseases [41].

In atherosclerosis, macrophage iron metabolism influences plaque formation and stability. Iron-loaded macrophages within atherosclerotic lesions produce ROS and pro-inflammatory cytokines, exacerbating oxidative stress and inflammation [41,42]. Additionally, iron retention in macrophages contributes to foam cell formation and plaque instability. Therapies aimed at reducing macrophage iron overload or enhancing iron efflux may offer novel approaches to preventing or treating cardiovascular diseases.

Iron dysregulation in macrophages is also implicated in neurodegenerative disorders, such as Alzheimer’s disease and Parkinson’s disease. Microglia, the macrophages of the central nervous system, accumulate iron in response to tissue damage and aging. Iron-loaded microglia produce neurotoxic ROS and pro-inflammatory cytokines, which contribute to neuronal damage and disease progression [43]. Understanding the interplay between iron metabolism and neuroinflammation may lead to new therapeutic strategies for neurodegenerative diseases.

## 7. Macrophages in Disease

Macrophages are versatile immune cells that play pivotal roles in a wide range of diseases, acting as both mediators of inflammation and key regulators of tissue repair. In various pathological contexts, macrophages can promote or resolve disease progression, depending on their activation state and the signals they receive from the microenvironment. In conditions such as autoimmune diseases, they often contribute to chronic inflammation and tissue damage, while in cancer, cardiovascular diseases, infections, and metabolic disorders, they can either exacerbate pathology or support healing processes. The ability of macrophages to switch between different functional states makes them essential players in disease dynamics, and understanding their role is crucial for developing therapies that can manipulate their activity for therapeutic benefit.

Although it is impossible to list all diseases involving macrophages, we present major types of diseases where macrophages play a central role (Table 5). The table outlines different disease groups, provides examples, highlights the role of macrophages, lists associated proteins/cytokines, and suggests potential therapeutic targeting. Later in the chapter, we delve into more detail on specific diseases that gather the most attention in scientific literature.

### 7.1. Autoimmune Diseases Involving Macrophages

In autoimmune diseases, macrophages can play dual roles, acting as mediators of inflammation and as regulators of immune tolerance. Dysregulation in macrophage activation or polarization can contribute to the pathogenesis of various autoimmune conditions [44].

#### 7.1.1. Rheumatoid Arthritis (RA)

Rheumatoid arthritis is a chronic inflammatory disorder that can affect more than just one joint. In RA, macrophages infiltrate the synovium and drive inflammation through the secretion of pro-inflammatory mediators, including TNF-α, IL-1β, and IL-6. These cytokines stimulate synovial fibroblasts, leading to the production of MMPs, which degrade cartilage and bone. Furthermore, activated macrophages release ROS and nitrogen intermediates that exacerbate tissue damage [8].

Macrophages in RA exhibit both M1 and M2 phenotypes, with M1 macrophages promoting inflammation and M2 macrophages attempting to repair tissue damage. However, this repair often becomes pathological, contributing to pannus formation and fibrosis. Emerging therapies targeting macrophages in RA include TNF-α inhibitors [45], IL-6 receptor blockers [46], and small molecules that inhibit macrophage infiltration or polarization into the M1 phenotype [47].

#### 7.1.2. Multiple Sclerosis (MS)

Multiple sclerosis is a chronic disease of the central nervous system (CNS) resulting in damage to the myelin covers of nerves in the brain and spinal cord. MS is unpredictable and may cause various symptoms. During disease progression, blood–brain barrier disruption allows macrophages to infiltrate the CNS, where they interact with microglia and other immune cells. M1 macrophages promote demyelination and axonal damage through the release of pro-inflammatory cytokines and ROS [48].

Conversely, M2 macrophages support repair by secreting anti-inflammatory cytokines, promoting remyelination, and clearing myelin debris. Therapies aiming to shift macrophage phenotypes from M1 to M2 have shown promise in experimental autoimmune encephalomyelitis (EAE), a mouse model of MS. For example, adoptive transfer of M2 macrophages or administration of M2-polarizing agents like IL-4 reduces disease severity and promotes repair [49].

#### 7.1.3. Systemic Lupus Erythematosus (SLE)

Systemic lupus erythematosus is an autoimmune disorder characterized by the development of antibodies against nuclear and cytoplasmic antigens, multisystem inflammation and a relapsing and remitting course. In SLE, impaired efferocytosis by macrophages leads to the accumulation of apoptotic cells and the release of intracellular components, including nucleic acids. These components serve as autoantigens, stimulating dendritic cells and B cells to produce autoantibodies. This process perpetuates a cycle of inflammation and tissue damage [50].

Macrophages in SLE are often polarized toward an inflammatory state, secreting cytokines such as type I interferons, IL-6, and TNF-α. Targeted therapies, including macrophage inhibitors [51] and efferocytosis-enhancing drugs [15], are under investigation to break this inflammatory cycle and restore immune tolerance.

#### 7.1.4. Type 1 Diabetes (T1D)

Type 1 Diabetes is a chronic disease of the pancreas affecting insulin production. In T1D, macrophages infiltrate pancreatic islets and contribute to the destruction of insulin-producing β-cells. They release pro-inflammatory cytokines, such as IL-1β, TNF-α, and IFN-γ, which induce β-cell apoptosis. The activation of M1 macrophages in the pancreas is a critical factor in initiating and sustaining insulitis, leading to the progression of T1D [52]. Studies have shown that these macrophages not only present antigens to autoreactive T cells but also exacerbate inflammation through cytokine production. Targeting macrophage activation and their inflammatory mediators is being explored as a therapeutic strategy to preserve β-cell function in T1D [53]. Promising approaches include phosphatidylserine (PS)-rich liposomes, which induce immunoregulatory changes in macrophages and reduce pro-inflammatory cytokine production [54]; dipeptidyl peptidase-4 inhibitors, ROS inhibitors (e.g., N-acetyl cysteine), and PPARβ/δ agonists (GW501516) to repolarize macrophages from M1 to M2 phenotype [55]; and disrupting type I interferon (IFN-I) signaling to limit autoreactive T-cell infiltration into pancreatic islets [56]. Additionally, macrophage depletion methods, such as clodronate-mediated depletion, have demonstrated efficacy in aborting disease progression in experimental models [56].

#### 7.1.5. Psoriasis

Psoriasis is a chronic autoimmune skin disorder characterized by hyperproliferation of keratinocytes and inflammation. Macrophages in psoriatic lesions are predominantly of the M1 phenotype, producing high levels of pro-inflammatory cytokines like TNF-α, interleukin-23 (IL-23), and IL-1β [57]. These cytokines promote the differentiation and activation of Th17 cells, which are central to the pathogenesis of psoriasis. Additionally, macrophage-derived cytokines contribute to the recruitment of other immune cells to the skin, amplifying the inflammatory response. Therapies targeting TNF-α and IL-23 have been effective in managing psoriasis [58], highlighting the essential role of macrophages in this disease.

#### 7.1.6. Crohn’s Disease

In Crohn’s disease, a type of inflammatory bowel disease, macrophages are key players in the chronic inflammation of the gastrointestinal tract. They exhibit an altered phenotype, leading to impaired bacterial clearance and excessive production of pro-inflammatory cytokines such as TNF-α and IL-6 [59]. This dysregulation contributes to tissue damage and the formation of granulomas, a hallmark of Crohn’s disease. Recent studies have implicated the NLRP3 inflammasome in macrophages as a contributor to the pathogenesis of Crohn’s disease by promoting IL-1β production and sustaining intestinal inflammation [60]. Modulating macrophage function and inhibiting specific inflammasome pathways are potential therapeutic approaches under investigation.

#### 7.1.7. Hashimoto’s Thyroiditis

Hashimoto’s thyroiditis is an autoimmune disorder characterized by the destruction of thyroid tissue, leading to hypothyroidism. Macrophages infiltrate the thyroid gland and contribute to tissue damage through antigen presentation and cytokine secretion. Th1-polarized immune responses dominate in this disease [61], with macrophages producing cytokines like IL-12 and IFN-γ, which further activate cytotoxic T cells against thyroid follicular cells. The interaction between macrophages and T cells perpetuates the autoimmune response, resulting in progressive thyroid dysfunction [62]. Understanding the role of macrophages in this context has led to interest in therapies that modulate macrophage activation to preserve thyroid function. Recent studies have highlighted several promising targets, including saikosaponin-d (SSd), which promotes anti-inflammatory M2 macrophage polarization [61]; hexokinase 3 (HK3), whose inhibition reduces pro-inflammatory M1 macrophage activation [63]; IL-1β signaling, which mediates thyrocyte destruction [64]; and macrophage migration inhibitory factor (MIF), which correlates with Th17-mediated thyroid damage [65].

#### 7.1.8. Systemic Sclerosis (Scleroderma, Ssc)

Systemic sclerosis is an autoimmune disease characterized by fibrosis of the skin and internal organs. Macrophages contribute to the fibrotic process by secreting pro-fibrotic cytokines such as TGF-β and IL-13, which stimulate fibroblast activation and extracellular matrix production [66]. Additionally, macrophages produce ROS and other mediators that promote tissue remodeling and fibrosis. The presence of alternatively activated (M2) macrophages in affected tissues correlates with disease severity and progression [67]. Targeting macrophage polarization and its pro-fibrotic activities is being explored as a therapeutic strategy to mitigate fibrosis in Ssc. Specific interventions include suppression of Sart1 to reduce M2 macrophage infiltration [68], modulation of NF-κB and JAK/STAT pathways to control inflammatory responses and macrophage polarization, and targeting the TGF-β/Smad pathway to attenuate fibrosis-associated M2 polarization [69]. Emerging therapies such as Pirfenidone and Janus kinase inhibitors further demonstrate promise by downregulating M2 macrophage polarization and associated fibrotic signaling [70]. Collectively, these strategies highlight macrophage polarization as a central therapeutic target in SSc-related fibrosis.

### 7.2. Non-Autoimmune Diseases Involving Macrophages

In non-autoimmune diseases, macrophages also play crucial roles, but their functions extend beyond immune regulation to include contributions to tissue damage, repair, and chronic inflammation. Depending on the context of the disease, macrophages can either exacerbate pathological processes or promote healing. In diseases such as cancer, cardiovascular conditions, infections, and metabolic disorders, the plasticity of macrophages allows them to adapt to changing microenvironments, influencing disease progression. Dysregulation in macrophage activation or polarization often leads to exacerbated inflammation, tissue damage, and poor outcomes, making macrophages a potential therapeutic target in these diseases.

#### 7.2.1. Infectious Diseases

In the immune response against infections, macrophages serve as both defenders that detect and eliminate pathogens and, paradoxically, as reservoirs that some pathogens exploit for survival. Their capacity to recognize microbial threats, initiate inflammation, and coordinate adaptive immune responses makes them key players in infection control. However, many pathogens have evolved mechanisms to manipulate macrophages, facilitating chronic infection and immune evasion.

Macrophages are equipped with a diverse array of PRRs that enable them to detect pathogens by recognizing conserved molecular patterns, known as pathogen-associated molecular patterns (PAMPs). PRRs such as toll-like receptors (TLRs), C-type lectin receptors (CLRs), and NOD-like receptors (NLRs) are critical for sensing bacterial, viral, fungal, and parasitic infections. Upon activation, these receptors initiate signaling cascades that lead to the production of pro-inflammatory cytokines, chemokines, and antimicrobial molecules [71].

For example, in tuberculosis caused by *Mycobacterium tuberculosis*, TLR2 and TLR4 on macrophages recognize components of the bacterial cell wall, such as lipoproteins and mycolic acids. This recognition triggers the nuclear factor kappa-light-chain-enhancer of activated B cells (NF-κB) signaling pathway, resulting in the production of IL-12 and TNF-α, which recruit and activate other immune cells. Additionally, NOD2 receptors detect bacterial peptidoglycans, amplifying the immune response [72].

In viral infections such as HIV, macrophages use TLR7 and TLR8 to detect single-stranded RNA, a key component of many viruses. These interactions trigger the production of interferon-alpha (IFN-α) and other antiviral cytokines, enhancing the cell’s ability to inhibit viral replication. Furthermore, macrophages release chemokines, such as CCL3 and CCL4, which attract natural killer (NK) cells and cytotoxic T lymphocytes to the site of infection [73].

Despite their robust detection mechanisms, macrophages must balance pathogen clearance with the need to avoid excessive inflammation that can damage host tissues. This balance is maintained through negative feedback loops involving anti-inflammatory cytokines like IL-10 and regulatory signaling pathways, such as the IRAK-M pathway.

While macrophages are highly effective at detecting and responding to pathogens, many microbes have evolved sophisticated strategies to evade their defenses and exploit them as niches for replication. *Mycobacterium tuberculosis* is a well-documented example, using multiple mechanisms to survive within macrophages. After being phagocytosed, *M. tuberculosis* inhibits phagosome–lysosome fusion, allowing it to persist in the nutrient-rich phagosomal compartment. The bacteria also produce proteins that neutralize ROS and reactive nitrogen species (RNS), key antimicrobial molecules produced by macrophages. This enables the bacteria to establish a long-term intracellular reservoir, leading to latent infection [72].

In HIV infection, macrophages serve as viral reservoirs due to their ability to harbor the virus for extended periods, even during antiretroviral therapy. HIV exploits macrophage entry receptors such as CD4 and CCR5 to infect the cells. Once inside, the virus integrates its genome into the host DNA, creating a latent reservoir that evades immune detection. Macrophages also contribute to viral dissemination by releasing infectious virions into surrounding tissues, perpetuating the cycle of infection [73].

Other pathogens, such as *Salmonella enterica*, utilize macrophages for intracellular replication. After being engulfed, *Salmonella* modifies the phagosomal environment using a type III secretion system, preventing lysosomal fusion and creating a replicative niche. Similarly, *Leishmania* parasites survive by inhibiting oxidative burst responses and altering macrophage signaling pathways [74]. These strategies highlight the dual role of macrophages as both immune effectors and exploitable hosts.

#### 7.2.2. Atherosclerosis

Macrophages are central to the development of atherosclerosis, a condition characterized by the buildup of plaques in arterial walls. They ingest oxidized low-density lipoprotein (LDL) particles, transforming into foam cells that accumulate within the arterial intima. This accumulation leads to the formation of fatty streaks, the earliest lesions in atherosclerosis. The death of foam cells and the subsequent release of their contents contribute to plaque instability and potential rupture, leading to cardiovascular events [75]. Targeting macrophage lipid uptake and promoting cholesterol efflux are potential therapeutic approaches to mitigate atherosclerosis. Recent studies highlight strategies such as macrophage-specific ATP citrate lyase (Acly) inhibition, which stabilizes plaques by disrupting cholesterol biosynthesis and enhancing apoptotic cell clearance [76]. Nanomaterials like the HA-Fc/NP3ST nano-module have also shown promise by promoting cholesterol efflux specifically in diseased macrophages, reducing plaque size and lipid accumulation [77]. Additionally, targeting Trem2 to regulate foamy macrophage differentiation [78] and enhancing ABCA1-dependent cholesterol efflux pathways [79] represent novel avenues for improving plaque stability and reducing vascular inflammation.

#### 7.2.3. Chronic Obstructive Pulmonary Disease (COPD)

In COPD, a progressive lung disease, macrophages contribute to chronic inflammation and tissue damage. Exposure to cigarette smoke and other pollutants activates alveolar macrophages, leading to the release of pro-inflammatory cytokines, chemokines, and proteases. These mediators attract neutrophils and other immune cells to the lungs, perpetuating inflammation and contributing to the destruction of lung tissue. Additionally, macrophages in COPD patients often exhibit impaired phagocytic function, reducing their ability to clear pathogens and apoptotic cells, which can exacerbate disease progression [80].

#### 7.2.4. Obesity and Type 2 Diabetes

Macrophages are implicated in the chronic low-grade inflammation associated with metabolic disorders. In obesity, adipose tissue macrophages shift from an anti-inflammatory M2 phenotype to a pro-inflammatory M1 phenotype. These M1 macrophages secrete cytokines, such as TNF-α and IL-6, which interfere with insulin signaling pathways, contributing to insulin resistance and the development of type 2 diabetes [52,81]. Strategies to modulate macrophage polarization in adipose tissue are being explored as potential therapies to improve insulin sensitivity and reduce metabolic inflammation. Approaches include targeting transcription factors such as PPARγ with Rosiglitazone to promote anti-inflammatory M2 macrophage infiltration [82], and modulating pathways like PI3K/AKT [83] and cAMP-EPAC signaling [84] that regulate macrophage polarization. Metabolic interventions, such as enhancing CYP2J2-EETs-sEH signaling, inhibit pro-inflammatory macrophage recruitment and maintain M2 phenotypes [84]. Pharmacological treatments including Metformin, DPP4 inhibitors (Linagliptin, Sitagliptin), and SGLT2 inhibitors (Empagliflozin) as well as dietary supplementation with unsaturated fatty acids further highlight promising strategies for controlling adipose macrophage polarization to combat metabolic dysfunction [82].

#### 7.2.5. Hepatic Fibrosis

Hepatic fibrosis is a progressive liver condition characterized by excessive extracellular matrix deposition, leading to scarring and impaired liver function. Macrophages, particularly Kupffer cells, play a pivotal role in the development and progression of hepatic fibrosis. They contribute to inflammation and fibrosis through the secretion of pro-inflammatory cytokines such as TNF-α and IL-1β, which activate hepatic stellate cells (HSCs) and promote extracellular matrix production. Recent studies have highlighted the heterogeneity of liver macrophages, with distinct subpopulations exhibiting different roles in fibrosis progression and resolution. For instance, Ly-6C^high^ macrophages are associated with the progression of liver fibrosis, while Ly-6C^low^ macrophages are linked to the degradation of extracellular matrix components and regression of fibrosis. Understanding the complex interactions between macrophages and other liver cells is crucial for developing targeted therapies aimed at modulating macrophage activity to treat hepatic fibrosis [85].

#### 7.2.6. Allergic Asthma

Macrophages play a pivotal role in the pathogenesis of allergic asthma, influencing both inflammation and airway remodeling. Upon exposure to allergens, macrophages polarize into different phenotypes, notably M1 (pro-inflammatory) and M2 (anti-inflammatory). In allergic asthma, M2 macrophages are predominantly involved and contribute significantly to airway inflammation and remodeling. These cells are driven by cytokines such as IL-4 and IL-13, which promote their differentiation into the M2 phenotype, leading to tissue remodeling and fibrosis in the lungs [86]. Additionally, interleukin-33 (IL-33) is another key cytokine that plays a role in the polarization of macrophages through the ST2 receptor, further exacerbating the inflammatory response in asthma. *ORMDL3*, a gene linked to asthma susceptibility, also influences macrophage function; transgenic mice overexpressing ORMDL3 show increased IgE levels and enhanced immune cell recruitment, including macrophages. The interplay of these proteins—IL-4, IL-13, IL-33, and ORMDL3—highlights the complex mechanisms by which macrophages contribute to the pathophysiology of allergic asthma [87]. Understanding these processes offers potential therapeutic avenues aimed at modulating macrophage activity to reduce asthma symptoms and improve lung function.

### 7.3. Role of Macrophages in Cancer

Macrophages are central players in the tumor microenvironment (TME), where their plasticity and adaptability enable them to support or inhibit tumor progression depending on microenvironmental signals. Within the TME, macrophages predominantly adopt an M2-like phenotype, which fosters tumor progression through the secretion of pro-angiogenic factors, immunosuppressive cytokines, and extracellular matrix remodeling enzymes. VEGF secretion by TAMs enhances angiogenesis, while MMPs degrade the extracellular matrix, facilitating metastasis. TAMs also secrete IL-10 and TGF-β, which suppress cytotoxic T-cell responses, and express PD-L1, further contributing to immune evasion. In addition, CCL2-mediated recruitment of monocytes into the TME sustains the presence of TAMs, ensuring continued tumor support [88].

Beyond these well-characterized mechanisms, metabolic reprogramming has emerged as a key determinant of TAM function and immune evasion. In response to the nutrient-deprived and hypoxic TME, TAMs shift from glycolysis-dominant metabolism to oxidative phosphorylation (OXPHOS) and fatty acid oxidation (FAO), which favor an immunosuppressive phenotype [89]. Metabolites such as lactate, produced by tumor cells, can stabilize HIF-1α, reinforcing TAM polarization toward a tumor-promoting phenotype and sustaining an immunosuppressive TME [90]. Furthermore, the accumulation of kynurenine through IDO1 activity suppresses T-cell function, while enhanced glutamine metabolism in TAMs facilitates their adaptation to the hostile TME [88].

Epigenetic modifications also contribute to the maintenance of the TAM phenotype. Histone modifications via EZH2 suppress pro-inflammatory gene expression, while DNA methylation silences genes associated with antigen presentation and T-cell activation, reinforcing the immune-suppressive state of TAMs. Additionally, chromatin remodeling has been shown to support the establishment of a trained immunity-like state, rendering TAMs resistant to reprogramming toward an anti-tumor phenotype [90]. These findings underscore the necessity of targeting TAM metabolism and epigenetics in future therapeutic strategies.

In addition to these metabolic and epigenetic pathways, recent studies have emphasized the pentose phosphate pathway (PPP) as another critical metabolic regulator influencing macrophage function in the TME. Pharmacological inhibition of PPP, specifically targeting transketolase (TKT) with oxythiamine, reprograms macrophages toward a pro-inflammatory, anti-tumor phenotype [91]. This shift boosts macrophage-mediated phagocytosis of lymphoma cells and diminishes their pro-tumorigenic activities by modulating glycogen metabolism and altering the UDPG-Stat1-Irg1-itaconate signaling axis, a key pathway in macrophage polarization [91]. Therefore, metabolic interventions targeting PPP, particularly TKT/TKTL1, represent a promising immunotherapeutic strategy to enhance macrophage anti-tumor functions and improve cancer immunotherapy outcomes.

## 8. Conclusions

Macrophages are essential mediators of immunity, inflammation, and tissue repair, exhibiting remarkable plasticity that allows them to adapt to diverse physiological and pathological conditions. Their roles in autoimmune diseases and cancer highlight their dual nature as both protectors and contributors to disease progression. Advancing our understanding of macrophage biology and polarization will open new therapeutic avenues, enabling precision medicine approaches that modulate macrophage function to treat a wide range of diseases. Ongoing research into macrophage-targeted therapies, including reprogramming, depletion, and enhancement of efferocytosis, holds significant promise for improving patient outcomes.

## Figures and Tables

**Figure 1 ijms-26-02107-f001:**
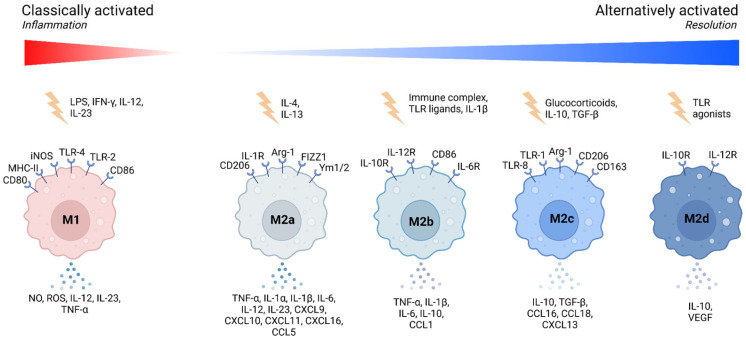
Macrophage polarization spectrum during inflammation. Picture created using BioRender.

**Figure 2 ijms-26-02107-f002:**
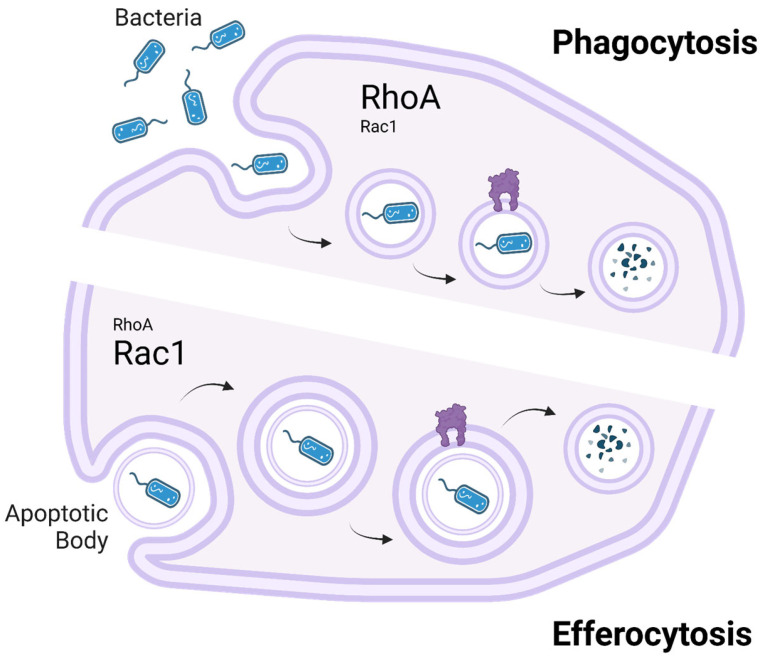
Comparison of phagocytosis and efferocytosis, highlighting the roles of RhoA and Rac1 in cytoskeletal regulation. Picture created using BioRender.

**Figure 3 ijms-26-02107-f003:**
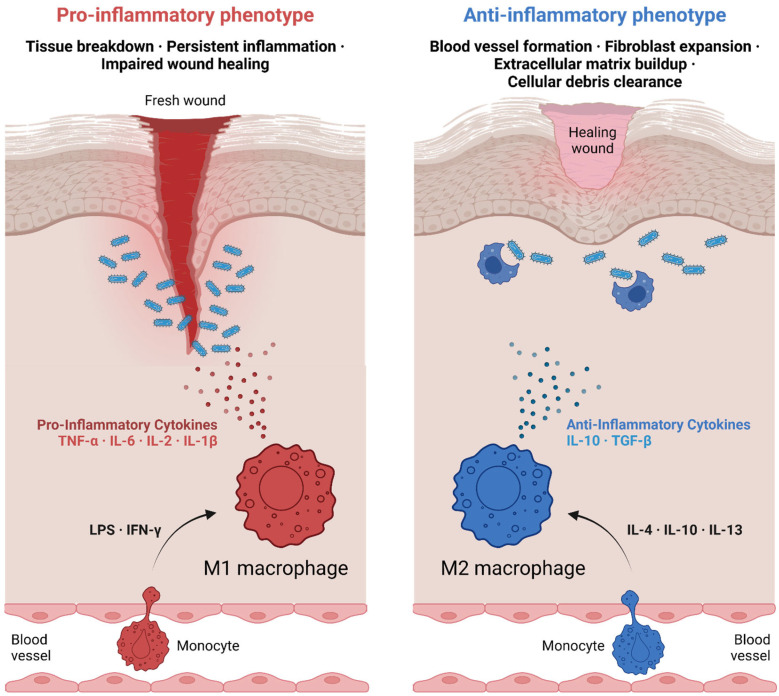
The role of macrophages in wound healing, from initial injury to tissue repair. Picture created using BioRender.

**Figure 4 ijms-26-02107-f004:**
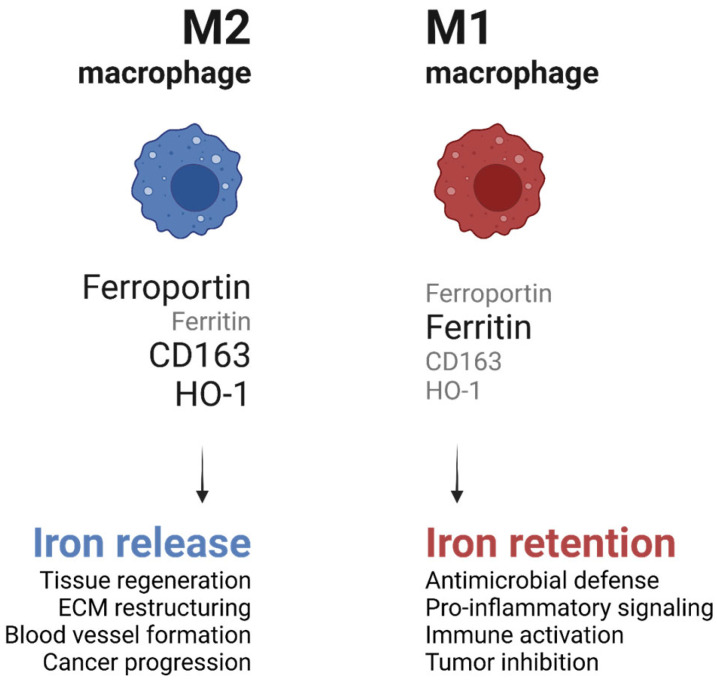
Role of M1 and M2 macrophages in iron homeostasis. Picture created using BioRender.

**Table 1 ijms-26-02107-t001:** Traditional activation pathway-based classification of macrophages.

Macrophage Type	Functions
Not activated	
M0 (Unpolarized)	Baseline surveillance of the tissue microenvironment; Basic phagocytosis of apoptotic cells and debris; Precursor state for M1 and M2 polarization; Maintains tissue homeostasis without initiating significant immune responses; High plasticity, capable of rapid reprogramming based on environmental signals.
Classically activated	
M1 (Pro-inflammatory)	Pathogen clearance (bacteria, viruses); Initiates and sustains pro-inflammatory responses; Produces ROS ^1^ and NO ^2^ to kill microbes; Tumoricidal activity; Promotes T-helper 1 (Th1) responses.
Alternatively activated	
M2a (Anti-inflammatory)	Tissue repair and regeneration; Promotes fibrosis and extracellular matrix remodeling; Produces anti-inflammatory cytokines (e.g., IL-10 ^3^, TGF-β ^4^); Angiogenesis.
M2b (Regulatory)	Balances pro- and anti-inflammatory responses; Modulates immune response through IL-10; Produces cytokines like IL-6 and TNF-α; Resolves inflammation.
M2c (Deactivated)	Promotes immune tolerance; Resolves inflammation; Tissue remodeling and repair; Reduces inflammation in chronic conditions.
M2d (Angiogenic and Tumor-Promoting)	Promotes tumor progression and metastasis; Induces angiogenesis through high VEGF production; Modulates the tumor microenvironment to suppress immune responses; Contributes to tissue remodeling and repair in hypoxic conditions.

^1^ Reactive oxygen species. ^2^ Nitric oxide. ^3^ Interleukin-10. ^4^ Transforming growth factor-beta.

**Table 2 ijms-26-02107-t002:** Examples of macrophage functions in specific microenvironments.

Macrophage Type	Functions
Tumor-Associated Macrophages (TAMs)	Supports tumor growth and metastasis; Promotes angiogenesis through VEGF ^1^; Suppresses immune responses via IL-10 and PD-L1 ^2^; Facilitates tumor immune evasion.
Foam Cells (Atherosclerosis)	Lipid uptake and storage; Chronic inflammation in atherosclerotic plaques; Produces pro-inflammatory cytokines and ROS; Contributes to plaque instability and progression.
Microglia (CNS Macrophages)	Neuroinflammation; Synaptic pruning in development; Secretes neurotrophic factors (e.g., BDNF ^3^); Removes damaged neurons and debris.
Kupffer Cells (Liver Macrophages)	Clearance of pathogens and endotoxins; Iron recycling through phagocytosis of aged RBCs ^4^; Produces complement factors for innate immunity; Regulates liver homeostasis and response to injury.
Wound-Healing Macrophages	Promotes angiogenesis and tissue repair; Secretes VEGF and TGF-β to rebuild tissues; Resolves inflammation in the wound site; Coordinates scar formation.
Osteoclasts (Bone Macrophages)	Bone resorption and remodeling; Maintains calcium homeostasis; Interacts with osteoblasts to regulate bone density.
Alveolar Macrophages	Clearance of pathogens and debris in the lungs; Regulates immune responses to airborne particles. Prevents unnecessary inflammation in alveoli.
Splenic Macrophages	Removes senescent RBCs and platelets; Processes iron recycling; Regulates immune surveillance in the spleen.
Peritoneal Macrophages	Maintains immune homeostasis in the peritoneum; Protects against infections in the abdominal cavity; Promotes resolution of inflammation.
Hofbauer Cells (Placental Macrophages)	Supports fetal development and maternal-fetal tolerance; Regulates placental inflammation; Protects against infections in the placenta.

^1^ Vascular endothelial growth factor. ^2^ Programmed death-ligand 1. ^3^ Brain-derived neurotrophic factor. ^4^ Red blood cells.

**Table 3 ijms-26-02107-t003:** Comparison of hybrid and specialized macrophage phenotypes.

Macrophage Phenotype	Inducing Factor	Key Characteristics	Primary Function
M4	CXCL4	Pro-inflammatory, foam cell-like properties, implicated in atherosclerosis	Atherosclerosis progression
Mox	Oxidative stress	NRF2-dependent transcriptional regulation, involved in tissue remodeling	Oxidative stress response, tissue remodeling
M(Hb)	Hemoglobin–haptoglobin complexes	Iron-recycling phenotype maintains some anti-inflammatory functions	Iron homeostasis, inflammation resolution
Mhem	Heme exposure	High HO-1 expression, promotes cholesterol efflux, reduces lipid accumulation	Plaque stability in atherosclerosis
Mres	Resolution phase of inflammation	Clears apoptotic cells, secretes anti-inflammatory mediators, supports tissue repair	Inflammation resolution, tissue repair

**Table 4 ijms-26-02107-t004:** Comparison of MHC-mediated antigen presentation by macrophages.

MHC Class	Antigen Source	Presented to	Function
MHC Class I	Intracellular antigens (e.g., viruses, tumor proteins)	CD8^+^ T cells (Cytotoxic T cells)	Initiates cytotoxic response against infected or malignant cells
MHC Class II	Extracellular antigens (e.g., bacterial peptides)	CD4^+^ T cells (Helper T cells)	Activates helper T cells to coordinate immune response
Cross-Presentation	Extracellular antigens presented on MHC I	CD8^+^ T cells	Enables immune response against tumors and viruses

**Table 5 ijms-26-02107-t005:** Types of diseases involving macrophages, with their primary (pathogenic) and secondary (regulatory) roles distinguished.

Disease Group	Examples	Role of Macrophages	Associated Proteins	Therapeutic Targeting
Pathogenic Role				
Autoimmune	Rheumatoid Arthritis, Systemic Lupus Erythematosus, Multiple Sclerosis, Inflammatory Bowel Disease, Psoriasis	Mediators of inflammation and tissue destruction	TNF-α, IL-1β, IL-6, IL-10	Inhibitors of pro-inflammatory cytokines (e.g., TNF inhibitors)
Fibrotic	Liver Fibrosis, Pulmonary Fibrosis, Cardiac Fibrosis	Promote fibrosis through secretion of profibrotic cytokines	TGF-β, IL-10, IL-4	Anti-fibrotic therapies, macrophage depletion
Hemophagocytic	Hemophagocytic Lymphohistiocytosis (HLH), Macrophage Activation Syndrome (MAS)	Hyperactivation leading to tissue damage and systemic inflammation	IL-6, TNF-α, IFN-γ	Immunosuppressive therapies, cytokine blockers
Regulatory Role				
Cancer	Breast Cancer, Lung Cancer, Colon Cancer	TAMs facilitate tumor growth	VEGF, IL-10, TGF-β, CSF-1 ^3^	Macrophage reprogramming, immune checkpoint inhibitors
Neurological	Alzheimer’s Disease, Parkinson’s Disease, Neuroinflammation	Modulate neuroinflammation, contribute to neurodegeneration	IL-1β, TNF-α, CCL2	Microglia/macrophage-targeted therapies
Pathogenic and/or Regulatory Role				
Infectious	Tuberculosis ^1^, HIV/AIDS ^1^, Leprosy ^1^, Malaria ^1^, Sepsis ^2^	Clearance of pathogens, but can cause excessive inflammation	IL-12, TNF-α, IL-6, TGF-β	Immunomodulators, pathogen-targeting therapies
Metabolic	Atherosclerosis ^1^, Type 2 Diabetes ^2^, Non-Alcoholic Fatty Liver Disease (NAFLD) ^2^	Chronic inflammation and insulin resistance	CCL2, IL-1β, TNF-α	Targeting macrophage polarization, anti-inflammatory agents
Chronic Inflammatory	Asthma, Chronic Obstructive Pulmonary Disease (COPD), Gout, Sarcoidosis	Airway inflammation ^1^, tissue remodeling ^2^, fibrosis ^1^	IL-4, IL-5 ^4^, IL-13, TNF-α	Bronchodilators, anti-inflammatory cytokine blockers

^1^ Macrophages play a pathogenic role. ^2^ Macrophages play a regulatory role. ^3^ Colony-stimulating factor 1. ^4^ Interleukin-5.

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
