# Peer review of "The Multifaceted Role of Macrophages in Biology and Diseases"

_ijms, 2025, doi:10.3390/ijms26052107_

Round 1
Reviewer 1 Report
Comments and Suggestions for Authors
A well-written macrophage biology review. Particularly, about macrophage plasticity's role in immune regulation and macrophages' involvement in several diseases. I have some minor comments:
1) In table 1 M2d are missing
2) Around line 78, more details about the receptors involved are needed. Specifically, specify that some receptors bind PS directly and others indirectly (Moon et al. 2023, EMM). BAI1 should be mentioned.
3) In Figure 2, what is the meaning of the font sizes of RhoA and Rac1. In the text, it says “Rac1 plays a more prominent role in both phagocytosis and efferocytosis…”
Reviewer 2 Report
Comments and Suggestions for Authors
This manuscript provides a comprehensive review of macrophage biology, emphasizing their polarization states, roles in autoimmune and inflammatory diseases, involvement in cancer progression, and potential as therapeutic targets. The review is well-structured and informative, presenting a balanced overview of macrophage functions in both health and disease. The use of tables and figures enhances readability, and the citations cover a broad range of relevant literature. However, several areas require further clarification, additional references, and minor structural improvements to enhance the overall impact and scientific rigor.
